# Jasmine Tea Attenuates Chronic Unpredictable Mild Stress-Induced Depressive-like Behavior in Rats via the Gut-Brain Axis

**DOI:** 10.3390/nu14010099

**Published:** 2021-12-27

**Authors:** Yangbo Zhang, Jianan Huang, Yifan Xiong, Xiangna Zhang, Yong Lin, Zhonghua Liu

**Affiliations:** 1Key Laboratory of Tea Science of Ministry of Education, Hunan Agricultural University, Changsha 410128, China; zhangyblucky@163.com (Y.Z.); jianan-huang@hotmail.com (J.H.); xyf951118@163.com (Y.X.); xynzzxn529@163.com (X.Z.); Yong-lin@hunau.edu.cn (Y.L.); 2National Research Center of Engineering and Technology for Utilization of Botanical Functional Ingredients, Hunan Agricultural University, Changsha 410128, China; 3Co-Innovation Center of Education Ministry for Utilization of Botanical Functional Ingredients, Hunan Agricultural University, Changsha 410128, China

**Keywords:** jasmine tea, CUMS, depression-like behaviors, neurotransmitters, gut microbiota

## Abstract

The number of depressed people has increased worldwide. Dysfunction of the gut microbiota has been closely related to depression. The mechanism by which jasmine tea ameliorates depression via the brain-gut-microbiome (BGM) axis remains unclear. Here, the effects of jasmine tea on rats with depressive-like symptoms via the gut microbiome were investigated. We first established a chronic unpredictable mild stress (CUMS) rat model to induce depressive symptoms and measured the changes in depression-related indicators. Simultaneously, the changes in gut microbiota were investigated by 16S rRNA sequencing. Jasmine tea treatment improved depressive-like behaviors and neurotransmitters in CUMS rats. Jasmine tea increased the gut microbiota diversity and richness of depressed rats induced by CUMS. Spearman’s analysis showed correlations between the differential microbiota (*Patescibacteria, Firmicutes, Bacteroidetes, Spirochaetes, Elusimicrobia,* and *Proteobacteria*) and depressive-related indicators (BDNF, GLP-1, and 5-HT in the hippocampus and cerebral cortex). Combined with the correlation analysis of gut microbiota, the result indicated that jasmine tea could attenuate depression in rats via the brain- gut-microbiome axis.

## 1. Introduction

Depression is a chronic disease threatening human health that is characterized by persistent low mood, slow thinking, cognitive impairment, dysphoria, loss of interest, and several other symptoms, ranging from psychomotor to cognitive impairments [1]. Depression is a highly prevalent mental illness worldwide, and people suffer from mild or severe depression [2]. The occurrence of depression is closely associated with neurotransmitter modulation. The dysfunction or damage of the neurotransmission system in the brain, especially involving 5-HT and BDNF, is intensively related to the neurobiological mechanisms of depression [3,4]. GLP-1 has an antioxidant and anti-inflammatory neuronal effect, which can increase striatal dopamine and catalepsy scores in a dose-dependent manner [5]. The increase in 5-HT, BDNF, and GLP-1 in depressive-like patients may help ameliorate depression. 

The microbiome plays an important role in depression [6,7]. Fecal microbiota transplantation from depressed patients to a rat whose microbiota is depleted could induce depression symptoms in the recipient animal, including anhedonia and anxiety-like behavior [8,9,10]. There is a bidirectional interaction between the microbiota and the brain [11,12]. It has been indicated that the onset of depressive disorder involves many aspects, mainly including the immune system, neurotransmitter, and hormonal system, and the activation of the brain-gut-microbiota mechanism [13,14]. Brain ischemia rapidly causes intestinal ischemia and increases excessive nitrate via free radical reactions, resulting in an imbalance of intestinal flora with the increase of *Enterobacteriaceae*, and the expansion of *Enterobacteriaceae* could enhance the systemic inflammation and exacerbate brain infarction [15].

The gut microbiota contains trillions of microorganisms, with more than 100 bacterial species, which can be mainly divided into six phyla: *Firmicutes* (such as *Clostridium, Enterococcus, Lactobacillus,* and *Ruminococcus*, which take up 60%), *Bacteroidetes* (such as *Bacteroides* and *Prevotella*, 15%), *Proteobacteria, Actinobacteria, Verrucomicrobia,* and *Fusobacteria* [16]. Depression is correlated with a decrease in the richness and diversity of the gut microbiota. It can be found that the compositions of gut microbiota in healthy people were significantly different from those of MDD patients according to the changes in the relative abundance of *Firmicutes, Actinobacteria,* and *Bacteroidetes* [12]. Traditional Chinese medicine (TCM) treatment could relieve psychiatric disorders by reducing the relative abundance of *Firmicutes* and *Ruminococcus;* increasing the abundance of *Bacteroidetes* and *Roseburia* [17].

The floral scent of jasmine tea was enhanced owing to tea adsorption of the volatiles released from *Jasminum sambac* flowers [18]. Jasmine tea is widely loved because of its aroma and physiological activity. The odor in jasmine tea could activate the parasympathetic nerve [19]. The root extract of *Jasminum sambac* has anti-inflammatory, analgesic, and antipyretic activities [20]. Addae et al. [21] found that the extract of jasmine leaf has a beneficial effect on an animal model of acute partial complex epilepsy, and has a positive anxiolytic effect at a dose that does not influent motor coordination.

It has been reported that jasmine tea could ameliorate the symptoms of depression. However, this mechanism is not yet clear. The brain-gut-microbiome axis plays a crucial role in the occurrence and treatment of depression. Therefore, we evaluated the effect of jasmine tea on CUMS-induced depression using behavioral tests and investigated the mechanism by analyzing the neurotransmitters and gut microorganisms in the present study. Compared with the published studies, two aspects of novel ideas were presented in the present study: (1) we evaluated the effect of jasmine tea on CUMS-induced depressive-like rats, not only focusing on the changes in behavioral tests but also aiming to explore the mechanism of attenuating-depression via the brain-gut-microbiome axis; (2) the present study reported the function of jasmine tea on the gut microbiota diversity and richness of depressed rats induced by CUMS.

## 2. Materials and Methods

### 2.1. Animals

Studies were performed on male Sprague Dawley rats purchased from Silaikejingda Experimental Animal Co., Ltd. (Changsha, China) (No. 43004700063334, (SCXK) 2016-002) at 4 weeks of age and housed in a specific pathogen-free unit under standard conditions (25 ± 1 °C; 55 ± 10% humidity; 12 h light/dark cycle) at Hunan Agricultural University. Rats were allowed to acclimatize for a minimum of 1 week before any experiments commenced. All experiments were conducted based on the Institutional Review Board and the Animal Care Committee ethical guidelines at Hunan Agricultural University (Changsha, China).

### 2.2. Induction of Depressive Rat Model

The procedure of CUMS was carried out according to Jing et al. [22] with slight modification. In short, a variety of mild stressors was contained in the CUMS processes: fasting and water deprivation for 20 h; water deprivation for 24 h; tilting squirrel cage 45° for 24 h; continuous light for 24 h; wet cage: 100 glitter + 200 mL water for 24 h; swimming at 4 ℃ for 5 min; horizontal shaking 5 min; behavior restriction for 2 h; tightness of a tail for 1 min. There were 9 stimuli in total, and one randomly or orderly was selected every day within 4 weeks. The procedural sequence was presented as follows (Figure 1): (1) stress procedure was performed during day 1 to day 28, (2) sugar preference test (SPT) was performed on day 29, (3) open-field test (OFT) was performed on day 30, and (3) forced swimming test (FST) was performed on day 31. In the behavioral tests, we transferred the rats to the laboratory room for adaptation at least 1 h for before the test. After the test, rats were sent back to their home cage. The rats in the control group were placed in a separate room to avoid contact with the stressed rats.

### 2.3. Experimental Design

After the acclimatization period, rats were randomly divided into 7 groups (*n* = 9). The doses of each group were presented as follows: Cons were given water (10 mL/kg) and served as the control group. Mods received the CUMS intervention for 4 weeks and were treated with water (10 mL/kg). Fluo received the CUMS intervention for 4 weeks and was treated with fluoxetine (1.82 mg/kg) (10 mL/kg). GM received the CUMS intervention for 4 weeks and was treated with green tea (64.8 mg/kg) (10 mL/kg). OL, OM, and OH received the CUMS intervention for 4 weeks and were treated combined with jasmine tea (10 mL/kg) at 21.6 mg/kg, 64.8 mg/kg, and 194.4 mg/kg, respectively. The soup was prepared each day with drinking water. Our jasmine tea was made with jasmine oil and green tea. The gavage dose of rats was determined according to the body weight of each rat. 

Feces of rats in each group at day 29 were collected and put into the liquid nitrogen immediately after collection, then stored at −80 °C until further analysis. 

### 2.4. Behavioral Test

The SPT, OFT, and FST were used to evaluate depressive-related behaviors. All behavioral experiments were tested by trained observers who are blinded to the treatments.

#### 2.4.1. SPT (Sugar Preference Test)

The consumption of sugar water reflects the degree of the animal’s response to the reward, and the consumption of sucrose water by the rats was used to simulate the human sense of interest. The SPT was performed as described by Mou et al. [23]. The sugar preferences were measured before grouping. First, rats were given 1% sucrose solution for 48 h. Then, rats were deprived of water for 24 h and then faced with two identical bottles (one bottle falling with 1% sucrose solution and the other with water) for 1 h. Then, the amount of liquid was recorded, and the consumption of sugar water was calculated. The sugar water consumption experiment was carried out again after 24 h of water shortages during the modeling process (day 7, day 14, day 21, and day 29).

Preference for sugar water (%) = consumption of sucrose water/(total amount of sucrose water + distilled water) × 100%.

#### 2.4.2. OFT (Open Field Test)

The OFT has been used to identify the mobility of animals. The OFT was performed as described by Mou et al. [23]. Rats were placed individually in a cardboard box (120 × 120 × 80 cm^3^), consisting of 16 squares (30 × 30 cm^2^) at the bottom. Briefly, the rat was separately put into the cardboard box. Placed the rat in the center of the box, and recorded the numbers of crossings (across the sector with all four paws) and standings (raising the forepaws) of each rat within 5 min after a 1 min habituation period. The cardboard box was cleaned between trials with a 95% ethanol solution.

#### 2.4.3. FST (Forced Swimming Test)

The FST has been used to identify depressive-like behavior in animals. The FST was performed as described by Jing et al. [22]. Briefly, the rat was separately put into a cylindrical glass container (80 cm height, 40 cm diameter) containing 60 cm of water (24 ± 0.5 °C). Care was taken not to put the nose of the rat below the water level. The rat was forced to swim for 5 min after a 1 min habituation period, and the immobility time (which was defined as no escape behavior, i.e., when they ceased struggling and remained floating motionless, only making those movements necessary to keep the head or nose above water.) was recorded during these 5 mins. After the test, the rat was removed from the water, immediately dried with towels, and returned to the cage.

### 2.5. Tissue Collection

After the end of all behavioral tests, rats were anesthetized with pentobarbital sodium and sacrificed after 12 h of fasting. The brain, colon, and serum samples were collected. A 4% paraformaldehyde solution was used to fix the colon for HE analysis. The hippocampus and cerebral cortex stripped out of the brain were put into the liquid nitrogen immediately after collection, then stored at −80 °C until further analysis.

### 2.6. Biochemical Analysis

The contents of BDNF, 5-HT, GLP-1 in the hippocampus and cerebral cortex were measured using rat BDNF, 5-HT, GLP-1 ELISA kits (Cusabio, Wuhan, China).

### 2.7. Histological Analysis and Morphometry

Histology analysis was carried out based on Liu’s method [24]. In brief, 4% paraformaldehyde-fixed colons were paraffin-embedded, sectioned at 3–6 μm, and stained with hematoxylin for histological analysis. Images were taken using a camera-equipped light microscope.

### 2.8. Intestinal Microbial Diversity Analysis

Analysis of gut microbiota was carried out according to Qian and Zhang et al. [25,26]. After stress stimulation, fresh feces of the rats were collected and frozen quickly in liquid nitrogen, and stored in a refrigerator at −80 °C for DNA extraction. In brief, microbial community genomic DNA was extracted from feces using the E.Z.N.A.^®^ soil DNA Kit (Omega Bio-Tek, Norcross, GA, USA). The concentration and purity of DNA were determined with a NanoDrop 2000 UV–vis spectrophotometer (Thermo Scientific, Wilmington, USA). The V3-V4 hypervariable region of the bacterial 16S rRNA gene was amplified with primers 338F (5′-ACTCCTACGGGAGGCAGCAG-3′) and 806R (5′-GGACTACHVGGGTWTCTAAT-3′) by an ABI GeneAmp^®^ 9700 PCR thermocycler (ABI, CA, USA). The PCR amplification was performed as follows: 3 min of denaturation at 95 °C, 27 cycles of 30 s at 95 °C, 30 s for annealing at 55 °C, and 45 s for elongation at 72 °C, and a final extension at 72 °C for 10 min, and a final extension at 4 °C. The PCR reactions were performed in triplicate 20 μL containing 4 μL 5 × TransStart FastPfu buffer, 2 μL 2.5 mM dNTPs, 0.8 μL forward primer (5 μM), 0.8 μL reverse primer (5 μM), 0.4 μL TransStart FastPfu DNA Polymerase, template DNA 10 ng, and ddH_2_O. The PCR product was extracted from a 2% agarose gel and purified using the AxyPrep DNA Gel Extraction Kit (Axygen Biosciences, Union City, CA, USA) and quantified using a Quantus™ Fluorometer (Promega, Madison, WI, USA).

#### 2.8.1. Illumina MiSeq Sequencing

Purified amplicons were pooled in equimolar amounts and paired-end sequenced (2 × 300) on an Illumina MiSeq platform (Illumina, San Diego, CA, USA) according to the standard protocols by Majorbio Bio-Pharm Technology Co. Ltd. (Shanghai, China). The raw reads were deposited into the NCBI Sequence Read Archive (SRA) database.

#### 2.8.2. Processing of Sequencing Data

The raw FASTQ files were demultiplexed using the Trimmomatic and FLASH based on their unique barcodes. Truncate the 300 bp reads at more than three sequential sites, the average quality score < 20 was accepted. Reads shorter than 50 bp containing barcode/primer errors or ambiguous base calls were discarded. Operational taxonomic units (OTUs) with 97% similarity cutoff were clustered using UPARSE (version 7.1, http://drive5.com/uparse/, 5 May 2020). In addition, the diversity and richness index, including Sobs, Ace, Shannon, and coverage indexes, were calculated to estimate the microbial diversity within an individual sample. Next, partial least squares discrimination analysis (PLS-DA) and hierarchical cluster analysis were performed by the Bray−Curtis distance matrix and average method by QIIME (version 1.7.0) and R software (version 2.15.3). The key bacterial taxa responsible for discrimination among the seven groups were identified with linear discriminant analysis effect size (LEfSe) analysis by calculating the effect of the abundance of each genus (LDA > 2.0 and *p*-value < 0.05).

### 2.9. Data Analysis

All data are shown as the mean ± standard error of the mean (SEM). Analyzed the data by ANOVA followed by Bonferroni’s test, and presented the graphics using GraphPad Prism software (8.0.2) to determine the significant difference between each group. A value of *p* < 0.05 was indicated significant.

## 3. Results

### 3.1. Jasmine Tea Reduced CUMS-Induced Depression and Ameliorated Depression-Like Behavior

Less body-gain and food-intake were observed in rats induced by CUMS than in rats without any treatment. Those depressive-like rats gained weight and food intake after jasmine tea intervention when compared with Mod (Figure 2A,B). The sugar water preference of rats in Mod was significantly decreased when compared with Con, which was significantly increased after treatment with jasmine tea (Figure 2C). The immobility time is a signal that rats had depressive-like symptoms in the FST. The results showed that rats in the Mod group had a significantly longer immobility time than those in the Con group (Figure 2D). Rats in CUMS could shorten the immobility time in the FST after being given different doses of jasmine tea soup (Figure 2D). The immobility time of CUMS rats in the FST was shortened after gavage with low doses of jasmine tea soups compared with middle and high doses. The behavioral activities and the ability to explore unknown things of depressive-like rats declined in the OFT after treatment with CUMS (Figure 2E). Jasmine tea consumption could decrease depressive symptoms in the OFT.

### 3.2. Jasmine Tea Increased Neurotransmitters in CUMS-Induced Depression

BDNF plays an important role in maintaining cell survival, synaptic plasticity, and neurotransmitter transmission, and an increase in BDNF improves brain cell viability and neurogenesis [27]. The levels of BDNF in the hippocampus in the Mod group were significantly decreased compared with those in the Con group and were significantly increased after jasmine tea intervention. The contents of BDNF in the OL group were the highest compared with those in the OM and OH groups (Figure 3A). GLP-1 effects extend beyond hypoglycemia because of its antioxidant and anti-inflammatory properties [5]. The dysfunction or impairment of 5-HT neurotransmission is intensively implicated in the neurobiological mechanisms of depression [3]. The levels of GLP-1 and 5-HT in the hippocampus and cerebral cortex of CUMS rats were significantly decreased compared with those without any external stimulus (Figure 3C–F). However, the levels of GLP-1 and 5-HT in depressed rats induced by CUMS were significantly increased after administration of jasmine tea soup compared with Mod, especially with the low dose of jasmine tea soup (Figure 3C–F). Together, these results suggested that jasmine tea soup, especially at low doses, could effectively ameliorate the symptoms of depression induced by CUMS via an increase in BDNF, GLP-1, and 5-HT.

### 3.3. Jasmine Tea Restored the Structure of Colon in CUMS-Induced Depression

The results of H&E staining examination in colon tissues are shown in Figure 4A–G. Compared with Con, Mod had a small amount of inflammatory cell infiltration in the colon tissues, and fewer goblet cells with scattered distribution, and shallow crypts. However, there were no obvious pathological changes in the structure of the colon among the Fluo, OL, and OM groups. The serosal, muscular, submucosa, mucosa layers, crypts, goblet cells, and large intestine glands were clearly visible in Fluo, OL, and OM groups. These results indicated that jasmine tea soup could ameliorate the dysfunction of colon tissue caused by CUMS and maintain the health of intestinal tissue.

### 3.4. Jasmine Tea Ameliorated the Gut Microbiota Communities in Rats with CUMS-Induced Depression

The gut microbiota plays an important role in the occurrence of depression and depression-associated diseases, which has been determined by using GF animals and microbiota transplants [28]. Recent evidence suggests that ameliorating the symptoms of depression by jasmine tea is associated with neurotransmitters. However, few studies have explored the mechanism by which jasmine tea ameliorates depression based on gut microbiota. To determine the function of the gut microbiota in mediating the beneficial effects of jasmine tea on CUMS-induced depression, rats were received CUMS for four consecutive weeks, resulting in symptoms of depression. At the same time, CUMS-induced rats were treated with different doses of jasmine tea soup. Subsequently, the effects of different doses of jasmine tea soup in CUMS and those rats without any external stimulus were compared.

The communities of gut microbiota showed that the diversity and richness of the microbial community were significantly reduced after external stimulus compared with the control group. The Shannon index results suggested that the diversity of microbial communities induced by CUMS was decreased, and the Sobs and Ace indexes showed that the richness of communities induced by CUMS was significantly lower than those of healthy rats (Figure 5). The diversity and richness of microbial communities changed by CUMS were significantly restored after the administration of jasmine tea.

To further explore the regulatory effect of the composition of the gut microbiota in jasmine tea alleviating CUMS-induced depression, different doses of jasmine tea on the microbial community in CUMS rats were evaluated. There is a clear separation between Con and Mod according to PLS-DA analysis (Figure 6A). The CUMS-induced depression rats were significantly changed after administration of jasmine tea, in which OL and OM were basically the same as Con (Figure 6A). The hierarchical cluster analysis at the genus level also demonstrated a significant separation induced by CUMS (Figure 7), which was divided into two groups: OM, OL, and Con; and GM, Mod, Fluo, and OH. Moreover, the taxonomic analysis showed that *Firmicutes* had a higher abundance in the gut microbiota of rats after CUMS-induced depression Figure 6B and Figure 7). These findings showed that the composition of gut microbiota was significantly reduced in CUMS-induced depressive rats and that these changes could be avoided after the administration of jasmine tea.

### 3.5. The Gut Microbiota Orchestrates CUMS-Induced Depression through Jasmine Tea

Treatment with jasmine tea significantly attenuated depression and improved microbiota community diversity and richness in CUMS-induced rats, which indicated that jasmine tea could ameliorate CUMS-induced depression via the gut microbiota. 

PLS-DA and hierarchical cluster analysis outlined the composition and structure of gut microbiota in rats (Figure 6A and Figure 7). 

Similar to other studies, *Firmicutes* and *Bacteroidetes* were the major components in Con (Figure 6B). After treatment with an external stimulus, there was an increase in *Firmicutes* (76.18%), and a decrease in *Bacteroidetes* (13.03%), resulting in a significant increase in the F/B ratio (584.65%) compared with Con (416.22%). The F/B ratio in CUMS rats was restored by jasmine tea intervention to 562.47%, 582.46%, and 471.13%, in OL, OM, and OH, respectively.

24 genera were identified as significantly different in the relative abundance of fecal microbiota in rats with versus without CUMS-induced depression (Con vs. Mod), including 10 genera increased and 14 genera decreased when compared with Con. CUMS-induced depression resulted in significant increases in the genera of *Ruminococcus_1, Lactobacillus, Blautia, Alloprevotella, Phascolarctobacterium, Clostridium_sensu_stricto_1, Romboutsia, Lachnoclostridium, Bacteroides, Coprococcus_2*; decreases in *norank_f__Muribaculaceae, unclassified_f__Lachnospiraceae, Akkermansia, Helicobacter, norank_f__Desulfovibrionaceae, norank_f__Ruminococcaceae, Desulfovibrio, Lachnospiraceae_NK4A136_group, Lachnospiraceae_UCG-010, Ruminiclostridium_9, Ruminococcaceae_UCG-014, norank_o__Mollicutes_RF39, Eubacterium]_coprostanoligenes_group, Ruminiclostridium_6* (Figure 7). However, the distribution of the abovementioned flora improved at a certain level after the intervention of jasmine tea, and the overall trend of the distribution was basically the same as that of Con (Figure 7).

On the one hand, ten kinds of gut microbiota at the genus level, including *norank_f__Muribaculaceae, unclassified_f__Lachnospiraceae, Candidatus_Stoquefichus, norank_f__Erysipelotrichaceae, Lachnospiraceae_UCG-010, Ruminococcaceae_NK4A214_group, Ruminiclostridium_9, Ruminococcaceae_UCG-014, norank_o__Mollicutes_RF39, and Eubacterium]_coprostanoligenes_group,* were significantly upregulated after jasmine tea treatment in rats with CUMS-induced depression (Figure 7). On the other hand, ten kinds of gut microbiota at the genus level, *Ruminococcus_1, Lactobacillus, Alloprevotella, Phascolarctobacterium, Clostridium_sensu_stricto_1, Romboutsia, Ruminococcus]_torques_group, Lachnoclostridium, Bacteroides,* and *Coprococcus_2*, were significantly downregulated after jasmine tea treatments in rats with CUMS-induced depression, especially at low and medium doses (Figure 7). Integratively, the analysis showed that jasmine tea could regulate depressive symptoms by downregulating *Lactobacillus,*
*Clostridium_sensu_stricto_1,* etc. These microbes were regarded as differential microorganisms in jasmine tea-treated groups (Figure 7).

To estimate the specific bacterial taxa associated with CUMS-induced depression or jasmine tea intervention, a LEfSe evolutionary branch diagram was presented among different treatments. The cladogram revealed that *Firmicutes, Proteobacteria, Actinobacteria, Bacteroidetes,* and *Tenericutes* at the phylum level were enriched significantly among each group. There were 19 genera with different taxonomic levels in the Con, including *g__unclassified_f__lachnospiraceae, g__ruminiclostridium_9, g__norank_f__ruminococcaceae, g__butyrivibrio,* etc., which belong to *Lachnospiracea, Ruminococcaeceae, Veillonellaceae, Corynebacteriaceae*. The model group had only one genus, *g__flavonifractor*. The enrichment and abundance of the bacteria were significantly increased after intervention with green tea dhool and jasmine tea, with 11 genera in GM, mainly including *c __Deltaproteobacteria, o__Desulfovibrionales, f__Desulfovibrionaceae, g__Desulfovibrio,* etc. The second group was OL, with nine genera, which mainly included *g__lachnospiraceae_UCG_010, g__Hydrogenoanaerobacterium, g__Anaerofilum, and g__Ruminiclostridium*. There were four genera in OH, including *o__Erysipelotrichales, f__Erysipelotrichaceae, c__Erysipelotrichia,* and *g__Candidatus_Stoquefichus*. There was only two enrichment flora in the OM group, *f__ruminococcus_, and g__ruminococcus_1*, and their LDA thresholds were relatively high, both of which were approximately 5 (Figure 6C,D). LEfSe Bar analysis further showed that the enrichment of intestinal microflora in rats was decreased after external stimulation, and depression symptoms could be alleviated by changing the enrichment of intestinal microflora after intervention with different doses of jasmine tea. 

Spearman’s rank correlations between the relative abundances of excellent gut microbiota and neurotransmitters were used to evaluate bacteria that might contribute to attenuating CUMS-induced depression. When considering the top 14 phyla or 50 genera in all seven groups, there were 6 phyla or 22 genera that had a Spearman’s correlation that was stronger than −0.5 or 0.5.

At the phylum level, 5-HT in the cerebral cortex was significantly correlated with *Patescibacteria, Firmicutes, Bacteroidetes, Spirochaetes,* and *Elusimicrobia*. *Patescibacteria* and *Bacteroidetes* were related to 5-HT in the hippocampus and BDNF in both the hippocampus and cerebral cortex. *Patescibacteria* and *Proteobacteria* were related to the levels of GLP-1 in the hippocampus and cerebral cortex (Figure 8). 

Those genera, including *Ruminiclostridium_6, Lachnospiraceae_UCG-010, Ruminiclostridium_5, Ruminococcaceae_UCG-013, norank_f__Muribaculaceae, Ruminococcaceae_UCG-014, Oscillibacter, Eubacterium]_coprostanoligenes_group, unclassified_f__Lachnospiraceae, Ruminiclostridium_9, norank_f__Ruminococcaceae, Streptococcus, norank_f__Desulfovibrionaceae, Ruminococcaceae_NK4A214_group, Lachnospiraceae_NK4A136_group, Aerococcus, Candidatus_Stoquefichus, Ruminococcus_2,* etc., showed a significant positive correlation with 5-HT, GLP-1, and BDNF in the hippocampus and cerebral cortex (Figure 9). Interestingly, these genera that had a positive correlation with neurotransmitter parameters were significantly increased by jasmine tea treatments. 

In summary, these findings revealed that the gut microbiota played a key role in regulating CUMS-induced depression after jasmine tea intervention. There may be a causal relationship between the relative abundances of gut microbiota and neurotransmitters (5-HT, GLP-1, and BDNF in the hippocampus and cerebral cortex).

We predicted functional composition profiles by performing phylogenetic reconstruction of unobserved states (PICRUSt) analysis and analyzed Kyoto Encyclopedia of Genes and Genomes (KEGG) level three categories of all samples among different groups based on 16S rRNA sequencing data. The results indicated that the abundance of metabolic pathways and biosynthesis of secondary metabolites were the most relevant among all KEGG pathways (Figure 10). 

## 4. Discussion

Depression is a major public health concern worldwide affecting numerous people [29]. The brain-gut-microbiome axis is an important biochemical link between the central nervous system (CNS) and enteric nervous system (ENS) [30]. The evidence has demonstrated that depression may be related to the gut microbiome [11,31,32]. The balance of the human intestinal microbiota will be disturbed by stress, and a series of subsequent mental health problems such as anxiety and depression will be caused [33]. The gut microbiome is an important and direct environmental contributor to the development of the central nervous system. It is composed of a larger number of bacterial and viral communities, which can significantly affect the health and disease of the host [34]. 

CUMS is an effective and reliable model that can simulate a variety of human depressive symptoms and major biochemical signs of depression, so it has been widely used to develop depression in animal models [22,23]. Jasmine tea could ameliorate depressive symptoms induced by CUMS [35]. However, few studies have shown whether jasmine tea attenuates the symptoms of depression via gut microbiota, and the mechanism of this process remains unclear. Therefore, in this study, the ameliorative depression effects with different doses of jasmine tea on CUMS-induced depressive rats were compared, and the orchestrated influences of gut microbiota and jasmine tea on CUMS-induced depression were investigated. The results indicated that jasmine tea relieved depression symptoms and that the diversity and richness of microbial communities changed by CUMS were significantly restored after being administered jasmine tea. 

The core symptom of depression is Anhedonia, which can be reflected by SPT and food consumption [25]. Jasmine tea-treated rats displayed significantly higher body weight, food intake, and sugar water preference than the model group, improving the anhedonia of depressive rats induced by CUMS. Jasmine tea treatment at low and medium doses also significantly shortened the immobility time in the FST and increased sugar preference in the SPT in the CUMS rats, suggesting that jasmine tea could ameliorate depressive behaviors. At the same time, jasmine tea intervention increased locomotor activity in the OFT.

BDNF mainly exists in the brain, and plays a major role in maintaining neuron survival, plasticity, neurogenesis, and synaptogenesis. Stress and depression reduced the expression and function of BDNF, and the antidepressants increased BDNF expression and blocked the growth factor expression deficits caused by stress and depression [4,27]. The quick improvement of BDNF contents in the hippocampus was responsible for the fast antidepressant-like effect [36,37]. In the present study, low and medium-dose jasmine tea improved the levels of BDNF in the hippocampus compared with the Mod group, indicating that the attenuation depression consequence of jasmine tea may be related to upregulation of the expression of BDNF.

GLP-1 is a kind of intestinal endocrine peptide, synthesized and secreted by intestinal endocrine L cells, and can enter brain tissue via the blood-brain barrier, and exert neuroprotection via the gut-brain axis [5,22]. In this study, the levels of GLP-1 in the cerebral cortex and hippocampus in CUMS rats were significantly reduced compared with those in the Con. Jasmine tea intervention raised the contents of GLP-1 in the cerebral cortex and hippocampus of depressive rats. We believe that GLP-1 in the brain might have a positive function in attenuating the depression caused by CUMS.

5-HT plays an important role in the feeling of well-being and happiness and can regulate mood, emotion, and behavior when stress occurs [38,39]. It has been indicated that there is a lower level of 5-HT in depressed patients compared with healthy people [3]. In this study, there was a significant reduction of 5-HT in the hippocampus and cerebral cortex of CUMS rats, while the levels in 5-HT CUMS rats were significantly increased after jasmine tea intervention, especially at the low and medium doses, which suggested that the attenuation of depression by jasmine tea might be mediated via the central monoaminergic neurotransmitter system. Above all, these results demonstrated that jasmine tea intervention effectively increased depressive-related neurotransmitters, including 5-HT, BDNF, and GLP-1 in the hippocampus and cerebral cortex.

Dietary supplements with tea had an energetic influence on preserving intestinal microecology [24]. Green tea polyphenols (GPTs) can boost mammalian energy conversion by regulating the structure of the gut microbial community, gene orthologs, and metabolic pathways [40]. In this study, Compared with Con, Mod had a small amount of inflammatory cell infiltration in the colon tissues, and fewer goblet cells with scattered distribution, and shallow crypts. However, there were no obvious pathological changes in the structure of the colon among the Fluo, OL, and OM groups. The serosal, muscular, submucosa, mucosa layers, crypts, goblet cells, and large intestine glands were clearly visible in Fluo, OL, and OM groups. These results showed that jasmine tea soup could ameliorate the symptoms of the colon induced by CUMS to tissue and maintain a healthy status of intestinal tissue.

The relationship between the imbalance of intestinal microbiota and depression can be mainly divided into three types. (1) Depression is related to tryptophan metabolism [8,41]. Tryptophan metabolism may be affected through microbiota by activating the enzyme indoleamine 2, 3-two plus oxygen (IDO) and the canine urine amino acid tryptophan to deplete tryptophan and serotonin [9]. (2) The intestinal flora can influence nutrient absorption and digestion, such as carbohydrates [42]. (3) Stresses, including psychosocial and psychophysical stress, can alter the gut flora, resulting in a decrease in *Lactobacillus* and *Bifidobacterium* populations, which is an important factor leading to depression [32,43].

However, it is still unclear whether the amelioration of depression affects the gut microbiota when treated with jasmine tea. To explore whether the richness and diversity of gut microbiota compositions would be affected after different doses of jasmine tea intervention, different doses of jasmine tea were administered to CUMS-induced depressive rats.

Similar to these studies [44], in this study, the results showed that the diversity and richness of gut microbiota in depressive rats induced by CUMS were significantly decreased compared with the Con, in which the relative abundance of *Firmicutes* in depressive rats increased and the relative abundance of *Bacteroidetes* decreased. The relative abundance of *Firmicutes* and *Bacteroidetes* changed after jasmine tea intervention. The results indicated that jasmine tea played a positive role in attenuating depression by changing the compositions of gut microbiota. The gut microbiota played an important role in amelioration CUMS-induced depression with jasmine tea.

Undoubtedly, CUMS-induced depression was closely related to the disturbance of the gut microbiota. Gut microbiome diversity has a strong association with mood-relating behaviors, including MDD, which has been characterized as a bidirectional community system between the brain and gut [45]. 

The richness and diversity of gut microbiota were decreased because of the mental induction of stress and depressive behavior in rodents [45]. To determine key bacteria and the underlying mechanisms of jasmine tea to reduce depression symptoms in a way that depends on the gut microbiota, the community diversity and structure of gut microbiota in CUMS-induced depressive rats with or without jasmine tea intervention were investigated. The results revealed that the diversity and richness of the gut microbiota community were decreased after CUMS stimulation. These results were consistent with previous findings, which indicated that decreased gut microbiota diversity was a characteristic of “depressive microbiota” [46]. Moreover, there was a higher community richness in low and medium dosages of jasmine tea. To reduce individual differentiation in the gut microecology of rats, we chose feces of rats induced by CUMS with or without jasmine tea administration as a research target in this study. Intriguingly, there were clear alterations in gut microbiota caused by CUMS based on the PLS-DA and hierarchical cluster analysis, which were reversed by low- and medium-dose jasmine tea treatment at a certain level, thereby indicating that intervention with low- and medium-dose jasmine tea had a positive influence on the gut microbiota of CUMS rats. The increase in the F/B ratio caused by CUMS was reduced by jasmine tea. An increase in the F/B ratio was thought to enhance energy harvesting and cause depression [44], and it was hypothesized that the positive effects of jasmine tea on CUMS-induced rats might be related to the active modulation of the F/B ratio. Grape extract (GE) could restore the dysbiosis of gut microbiota by increasing the observed species, changing the F/B ratio, and increasing the relative abundance of *Bifidobacteria, Akkermansia,* and *Clostridia* genera [47]. The abundances of *Lachnospiraceae* and *Ruminococcaceae* were also significantly decreased in depressed patients [45]. The increase in *Bacteroidetes* corresponded to a higher abundance of *Bacteroides* and *Parabacteroides,* and the lower abundance of other bacterial components after antidepressant treatment, while the decrease in *Firmicutes* was mainly due to the relatively low abundance of *Ruminococcaceae_UCG-014* [26]. Anatomizing the microorganisms, we found that microorganisms were significantly restored by jasmine tea and were significantly related to neurotransmitters, including *Patescibacteria, Firmicutes, Bacteroidetes, Spirochaetes, Elusimicrobia,* and *Proteobacteria*. *The genera, Ruminiclostridium_6, Lachnospiraceae_UCG-010, Ruminiclostridium_5, Ruminococcaceae_UCG-013, norank_f__Muribaculaceae, Ruminococcaceae_UCG-014, Oscillibacter, Eubacterium]_coprostanoligenes_group, unclassified_f__Lachnospiraceae, Ruminiclostridium_9, norank_f__Ruminococcaceae, Streptococcus, norank_f__Desulfovibrionaceae, Ruminococcaceae_NK4A214_group, Lachnospiraceae_NK4A136_group, Aerococcus, Candidatus_Stoquefichus, Ruminococcus_2,* etc., had a positive relationship with 5-HT, GLP-1, and BDNF in the hippocampus and cerebral cortex. These results indicated that attenuation of the symptoms of depression by jasmine tea was related to the gut microbiota and neurotransmitters in the hippocampus and cerebral cortex. 

The brain-gut axis is a bidirectional nodal axis that interacts with the brain and the gastrointestinal tract, which consists of the CNS, autonomic nervous system (ANS), ENS, HPA axis, etc., and the functions of each part are coordinated with each other [16,30]. Intestinal flora changes will cause the host inflammatory response, influence the absorption of nutrients, and change neurotransmitter metabolism and activity in the HPA axis and BDNF levels (lower BDNF levels in the hippocampus are related to depression and anxiety behavior), which will lead to nervous system disorders, such as depression [48]. *C. butyricum* significantly reduced cognitive dysfunction and histopathological changes in VaD mice, which is mainly reflected in the increase in BDNF and Bcl-2, the decrease in Bax, and induce Akt phosphorylation (p-Akt) and reduce neuronal apoptosis [49]. There is an interaction between the gut microbiota and CNS and ENS to regulate nutrient metabolism via enteroendocrine cells (EECs), such as cholecystokinin (CCK), GLP-1, 5-HT, and serotonin [16]. 5-HT can be produced by *Bacillus, Escherichia,* and *Saccharomyces*, while DA can be secreted by *Bacillus* and *Serratia* [16]. The relative fecal abundance of the *Bifidobacterium, Lactococcus,* and *Lactobacillus* genera in HFD (high-fat diet)-fed rats was increased by probiotics [50]. Probiotics can also ameliorate a range of behaviors related to depression in mice and rats [51]. Low levels of *Lactobacillus* and *Turicibacter* resulted in various disorders, including depression, and the intervention of certain species of *Lactobacillus* ameliorated depressive behaviors in animal models [52]. Fecal microbiota transplantation improved depressive-like behaviors, altered the imbalance in gut microbiota, and alleviated intestinal inflammation, intestinal mucosal destruction, and neuroinflammation in CUMS rats [28].

Many shreds of evidence have indicated that gut microbiota compositions have a closer association with host metabolism [53]. The gut microbiota plays a major role in regulating intestinal health via the metabolism of carbohydrates, lipids, and amino acids. The metabolism of depressive rats firstly showed disturbances of microbial genes and host metabolites, including carbohydrates (depressive mice need higher energy) and amino acid metabolism [12]. GTPs decreased colorific carbohydrates, such as glucose, fructose, and trehalose, in the experimental rats, and improved glycolysis and the metabolism of amino acids in the gut, in addition to promoting mitochondrial TCA cycle and urea cycle dependent on the gut microbiota [40]. 

Above all, these results illustrated that the diversity and richness of gut microbiota in depressed rats was significantly reduced with the interference of external chronic stress, and the diversity and richness of gut microbiota were improved when treated with the different doses of jasmine tea, which could alleviate the depressive symptoms of rats caused by CUMS. Taken together, these findings indicated that jasmine tea significantly ameliorated the symptoms in a CUMS model. However, the beneficial influences of green tea dhool at a medium dose had little effect on ameliorating the symptoms of depression when compared with the same dose of jasmine tea. Furthermore, jasmine tea intervention restored the diversity and richness of the gut microbial components in CUMS rats mainly through metabolic pathways and biosynthesis of secondary metabolites. We analyzed the effect of jasmine tea on the intestinal microorganisms of CUMS at the levels of BDNF, 5-HT, and GLP-1 in the hippocampus and cerebral cortex, and explored the possible mechanism by which jasmine tea ameliorates CUMS-induced depression in this study. However, the patronization mechanism of depression involves many aspects. In future research, we will combine a variety of research methods and perspectives to further investigate the mechanism by which jasmine tea attenuates depression caused by CUMS.

## 5. Conclusions

The purpose of this study was to explore whether jasmine tea could ameliorate depression-like symptoms induced by CUMS and its possible mechanism via the brain-gut-microbiome axis. Male rats were subjected to CUMS and treated with jasmine tea soup. Depressive-like behaviors were measured by a series of behavior tests. The contents of 5-HT, brain- BDNF, GLP-1, and gut microbiota were evaluated. Jasmine tea intervention significantly attenuated CUMS-induced depressive-like behavior in rats. Meanwhile, jasmine tea treatment exhibited significant effects, upregulating the expression of BDNF, 5-HT, and GLP-1 in the hippocampus and cerebral cortex. These neurotransmitters were correlated with *Patescibacteria, Firmicutes, Bacteroidetes, Spirochaetes, Elusimicrobia,* and *Proteobacteria*. Taken together, our results demonstrated that jasmine tea had a positive preventive effect on CUMS rats partially attributed to stimulation of cerebral BDNF, 5-HT, and GLP-1 and activation of the composition of gut microbiota through the metabolic pathway, biosynthesis of secondary metabolites, and biosynthesis of amino acids.

## Figures and Tables

**Figure 1 nutrients-14-00099-f001:**
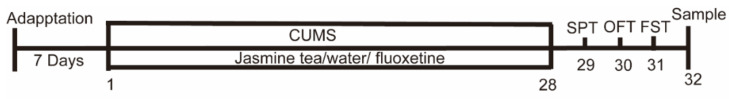
The schematic diagram of the experiment. CUMS Rats were treated with jasmine tea/green tea/water/fluoxetine during the 4 weeks. After 4 weeks of treatment, fresh feces were collected, and then a battery of behavioral tests relevant to depression was undergone. The rats were sacrificed on day 32. Tissues were collected to analysis in further. CUMS, chronic unpredictable mild stress; SPT, sugar preference test; OFT, open field test; FST, forced swim test.

**Figure 2 nutrients-14-00099-f002:**
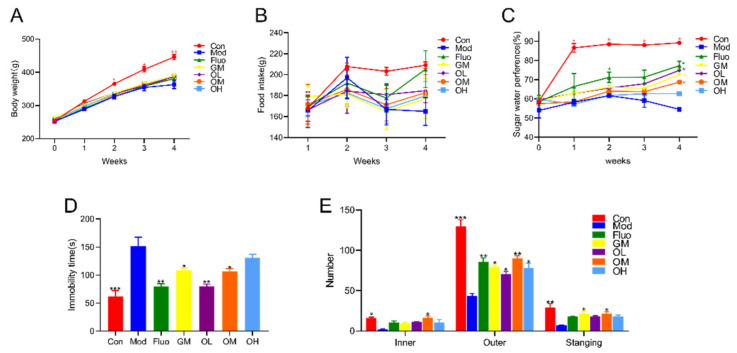
Jasmine tea ameliorated the depressive behavior induced by CUMS. (**A**). Body weight; (**B**). food intake; (**C**). sugar water preference; (**D**). immobility time in FST; (**E**). number in the OFT. Con, control group; Mod, CUMS group; Fluo, CUMS rats treated with fluoxetine; GM, CUMS rats treated with green tea dhool at a middle dose; OL, CUMS rats treated with jasmine tea at a low dose; OM, CUMS rats treated with jasmine tea at a middle dose; OH, CUMS rats treated at a jasmine tea with high dose. (* *p* < 0.05 vs. Mod, ** *p* < 0.01 vs. Mod, *** *p* < 0.001 vs. Mod).

**Figure 3 nutrients-14-00099-f003:**
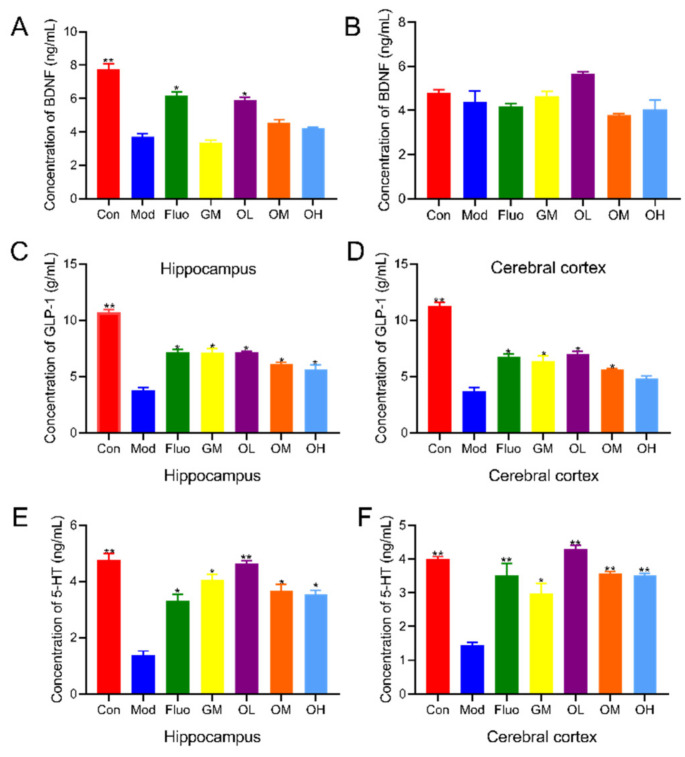
Jasmine tea ameliorated neurotransmitters in the hippocampus and cerebral cortex in depressive rats. (**A**). Concentration of BDNF in the hippocampus; (**B**). concentration of BDNF in the cerebral cortex; (**C**). concentration of GLP-1 in the hippocampus; (**D**). concentration of GLP-1 in the cerebral cortex; (**E**). concentration of 5-HT in the hippocampus; (**F**). concentration of 5-HT in the cerebral cortex. Con, control group; Mod, CUMS group; Fluo, CUMS rats treated with fluoxetine; GM, CUMS rats treated with green tea dhool at a middle dose; OL, CUMS rats treated with jasmine tea at a low dose; OM, CUMS rats treated with jasmine tea at a middle dose; OH, CUMS rats treated with jasmine tea at a high dose. (* *p* < 0.05 vs. Mod, ** *p* < 0.01 vs. Mod).

**Figure 4 nutrients-14-00099-f004:**
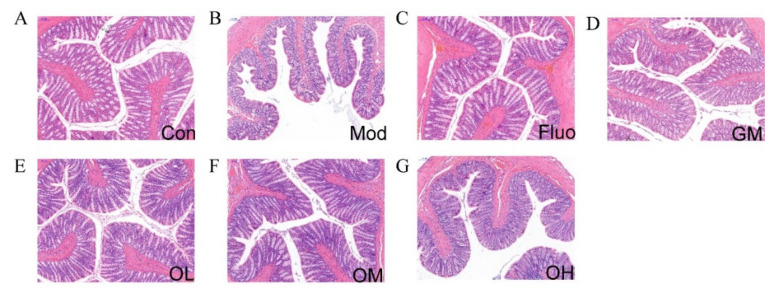
The result of colon HE staining. (**A**). Con, control group; (**B**). Mod, CUMS group; (**C**). Fluo, CUMS rats treated with fluoxetine; (**D**). GM, CUMS rats treated with green tea dhool at a middle dose; (**E**). OL, CUMS rats treated with jasmine tea at a low dose; (**F**). OM, CUMS rats treated with jasmine tea at a middle dose; (**G**). OH, CUMS rats treated with jasmine tea at a high dose.

**Figure 5 nutrients-14-00099-f005:**
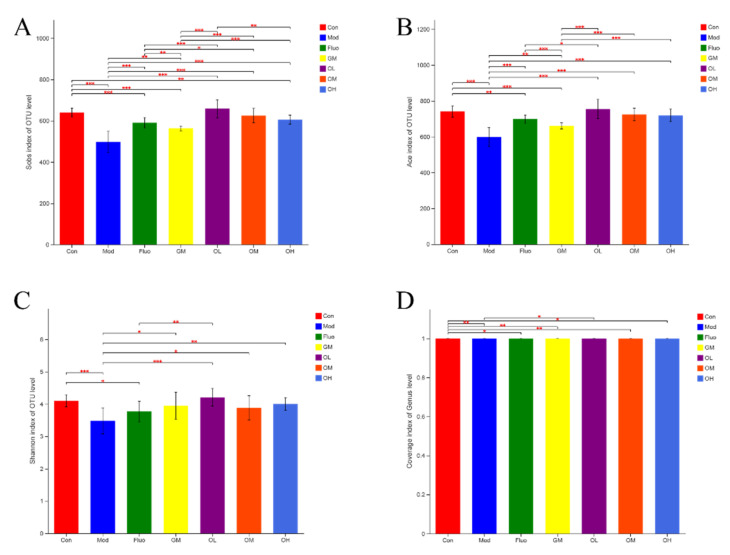
Comparison analysis of the difference in the α diversity index of gut microbiota among different groups. (**A**). Difference in Sobs index. (**B**). Difference in Ace index. (**C**). Difference in Shannon index. (**D**). Difference in Coverage index. (* *p* < 0.05, ** *p* < 0.01, *** *p* < 0.001).

**Figure 6 nutrients-14-00099-f006:**
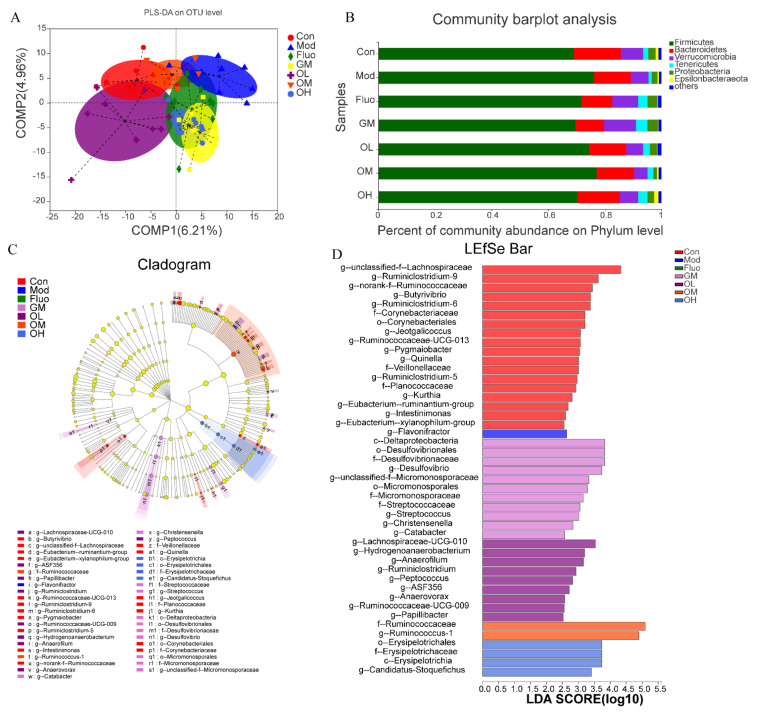
Jasmine tea modulated gut microbiota compositions of depressive-like rats. (**A**). PLS-DA analysis of gut microbiota based on OTU relative abundance among the control group and CUMS-induced depression or jasmine tea administration. (**B**). Taxonomic profiling of bacteria at the phylum level of gut microbiota in different treatments. (**C**). Cladogram analysis among different groups. The central point represents the root of the tree (bacteria), and each ring represents the next lower taxonomic level (phylum through OTUs). The diameter of each circle represents the relative abundance of the taxon. When full identification was not possible, g_ or s_ alone was used for the genus or species, respectively. (**D**). Column chart of linear discriminant analysis (LDA). PLS-DA, Partial least squares discrimination analysis.

**Figure 7 nutrients-14-00099-f007:**
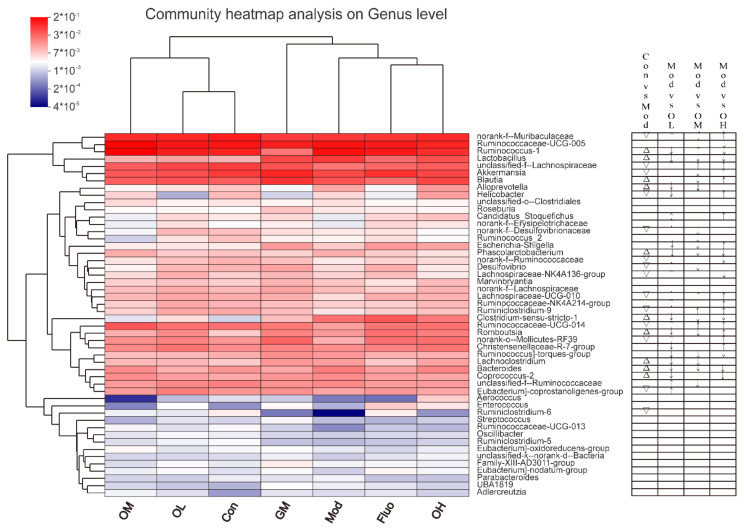
The compositions of gut microbiota were altered after jasmine tea administration. The relative abundance of the top 50 genera among different groups is shown in the heatmap. The relative abundances are presented by the color of the squares. The similarities of abundance among genera or samples are indicated by hierarchical clustering. The arrows (↑) and (↓) represent that the relative abundance of different genera was significantly up-regulated and down-regulated after the intervention of different doses of jasmine tea compared with the model group. The down-triangle (▽) and up-triangle (Δ) represent that the relative abundance of different genera was significantly down-regulated and up-regulated after the CUMS intervention compared with the control group.

**Figure 8 nutrients-14-00099-f008:**
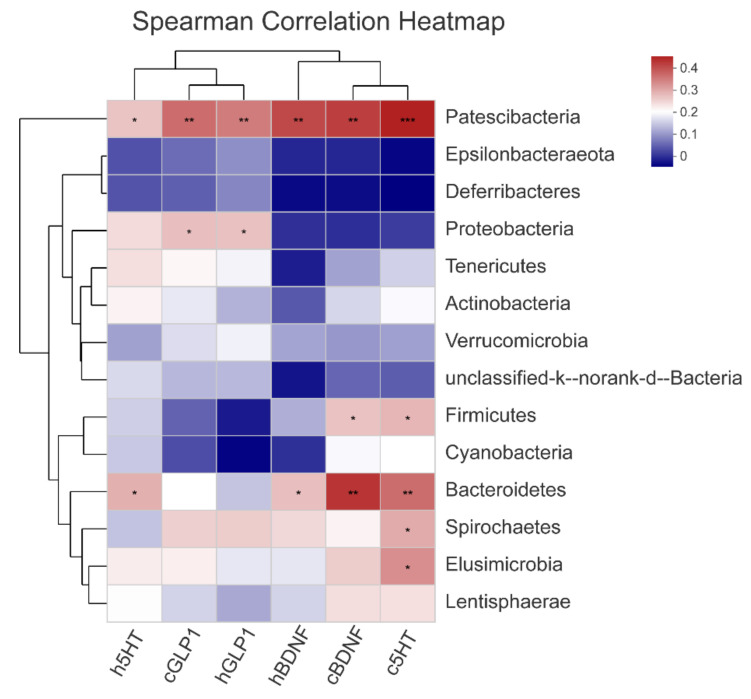
Correlations between the abundances of phyla and neurotransmitters in CUMS rats. The heatmap shows Spearman’s rank correlation between neurotransmitters and the relative abundances of the selected phyla. The most abundant phyla found in rats with at least one more extreme correlation than −0.5 or 0.5 are indicated. The asterisk (*) illustrates the significance of the correlations between neurotransmitters and the relative abundances of the phyla. Hierarchical clustering was presented for both phylum and neurotransmitters based on the Euclidean distances between Spearman’s rank correlations. (* *p* < 0.05, ** *p* < 0.01, *** *p* < 0.001).

**Figure 9 nutrients-14-00099-f009:**
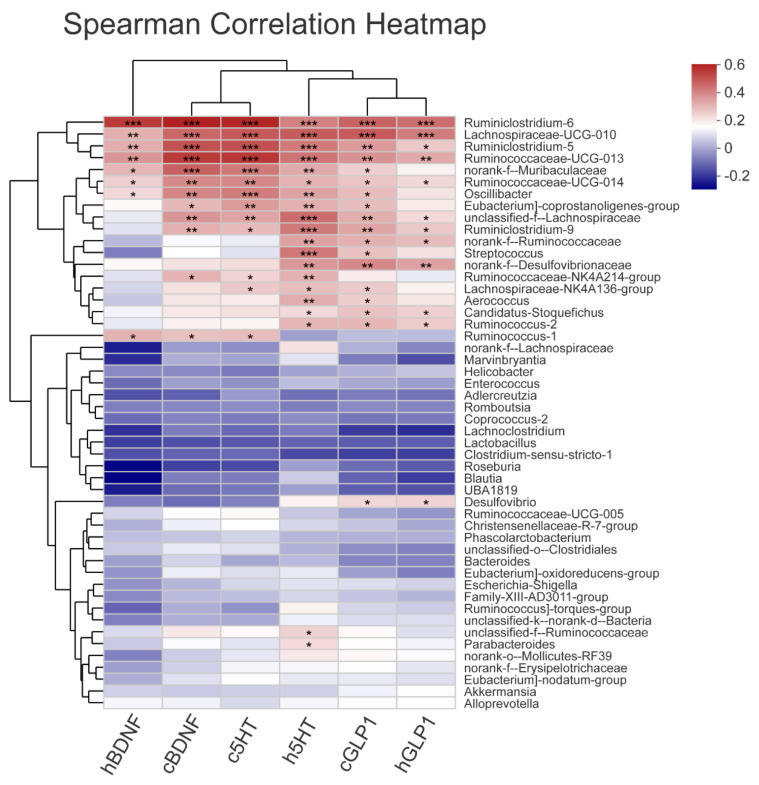
Correlations between the abundances of genera and neurotransmitters in CUMS rats. The heatmap shows Spearman’s rank correlation between neurotransmitters and the relative abundances of the selected genus. The most abundant rat genera with at least one more extreme correlation than −0.5 or 0.5 are indicated. The asterisk (*) illustrates the statistical significance of the correlations between neurotransmitters and the relative abundances of the genera. Hierarchical clustering was presented for both genera and neurotransmitters, based on the Euclidean distances between Spearman’s rank correlations. (* *p* < 0.05, ** *p* < 0.01, *** *p* < 0.001).

**Figure 10 nutrients-14-00099-f010:**
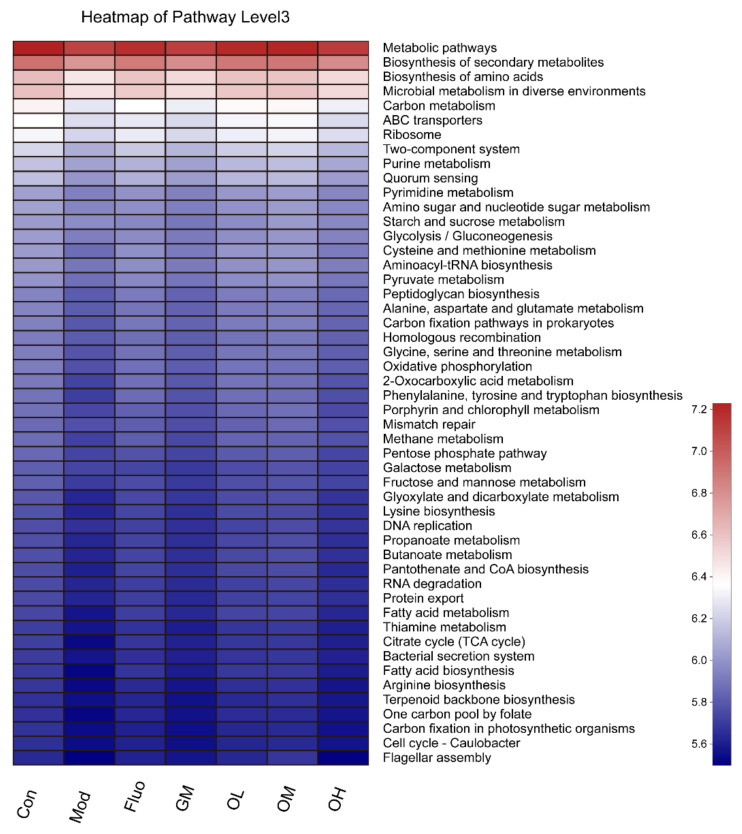
KEGG metabolic pathways enriched in the gut microbiota of CUMS rats.

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
