# Peer review of "Jasmine Tea Attenuates Chronic Unpredictable Mild Stress-Induced Depressive-like Behavior in Rats via the Gut-Brain Axis"

_nutrients, 2021, doi:10.3390/nu14010099_

Round 1

Reviewer 1 Report

  1. The abstract does not fully describe the research. It is not clear what do the authors mean by the term harmful microbiota. Also the authors describe the relationship between microbiota and depression symptoms. However it is not clear is the relationship represents positive or negative correlation.
  2. In section 2.8.2, it is not clear which group of mice have been used.
  3. figure 2 seems to be missing explanation of abbreviations
  4. Figure 4 description is not clear, can you highlight points of inflammation ?
  5. Section 3.5, the effect of depression on the microbiota and the effect of Jasmine tea is clear
  6. Figure 7 is not clear even in higher magnification, specifically the legend of the species and the table of abundance  
  7. Proof-reading might be needed.

Author Response

Response to Reviewer Comments

Dear Reviewers:

Thank you for your letter and for the reviewers’ comments concerning our manuscript entitled “Jasmine Tea Attenuates Chronic Unpredictable Mild Stress-induced Depressive-Like Behavior in Rats Via the Gut-Brain Axis”. Those comments are all valuable and very helpful for revising and improving our paper. We have seriously taken the comments into consideration in revising our manuscript. Revised portion are marked in red in our manuscript. The main corrections in the paper and the responds to comments are as follows:

Reviewer #1: 1. The abstract does not fully describe the research. It is not clear what do the authors mean by the term harmful microbiota. Also, the authors describe the relationship between microbiota and depression symptoms. However, it is not clear is the relationship represents positive or negative correlation.

Response: Thank you for your reasonable suggestion. The harmful microbiota in this term means those microbiomes, including, Ruminococcus_1, Blautia, Alloprevotella, Phascolarctobacterium, Clostridium_sensu_stricto_1, Romboutsia, Lachnoclostridium, Bacteroides, Coprococcus_2, played a positive role in depression occurrence. The relationship between the relative abundance of dominant microbiome and neurotransmitters were analyzed by Spearman’s rank correlations. At the phylum level, 5-HT in the cerebral cortex was significantly correlated with Patescibacteria, Firmicutes, Bacteroidetes, Spirochaetes, and Elusimicrobia. Patescibacteria and Bacteroidetes were related with 5-HT in hippocampus and BDNF in both hippocampus and cerebral cortex. Patescibacteria and Proteobacteria were related to the levels of GLP-1 in hippocampus and cerebral cortex. Those genera, including Ruminiclostridium_6, Lachnospiraceae_UCG-010, Rumini-clostridium_5, Ruminococcaceae_UCG-013, norank_f__Muribaculaceae, Ruminococca-ceae_UCG-014, Oscillibacter, Eubacterium]_coprostanoligenes_group, unclassi-fied_f__Lachnospiraceae, Ruminiclostridium_9, norank_f__Ruminococcaceae, Streptococcus, norank_f__Desulfovibrionaceae, Ruminococcaceae_NK4A214_group, Lachnospirace-ae_NK4A136_group, Aerococcus, Candidatus_Stoquefichus, Ruminococcus_2, etc. showed a significant positive correlation with 5-HT, GLP-1 and BDNF in the hippocampus and cerebral cortex

2.In section 2.8.2, it is not clear which group of mice have been used.

Response: The group of rats in this section were included in the control group, the CUMS model group, the Fluo group, CUMS rats treated with fluoxetine; the GM group, CUMS rats treated with green tea dhool with middle dose; the OL group, CUMS rats treated with jasmine tea with low dose; the OM group, CUMS rats treated with jasmine tea with middle dose; the OH group, CUMS rats treated with jasmine tea with high dose.

3.figure 2 seems to be missing explanation of abbreviations

Response: We fully agree with your reasonable suggestion. The explanation of abbreviations of figure 2 and figure 3 had added in the manuscript.

4.Figure 4 description is not clear, can you highlight points of inflammation?

Response: As illustrated in Figure 4A, the colon mucosa in the control group was intact, epithelial cells were arranged neatly and there was no infiltration by inflammatory cell.

Compared with Con, the structure of colon in Mod had a small amount of inflammatory cell infiltration in the colon tissues, fewer goblet cells with scattered distribution, and shallow crypts.

5.Section 3.5, the effect of depression on the microbiota and the effect of Jasmine tea is clear

Response: The diversity and richness of the microbial community were analyzed by the α diversity. The results showed that the Shannon index suggested that the diversity of microbial communities induced by CUMS was decreased, and, the Sobs and Ace indexes showed that the richness of communities induced by CUMS was significantly lower than those healthy rats. The results indicated that the diversity and richness was reduced by CUMS-induced depression. However, the Shannon index, Sobs and Ace indexes was increased after jasmine tea intervention.

6.Figure 7 is not clear even in higher magnification, specifically the legend of the species and the table of abundance  

Response: We had modified those figures in the manuscript by adjusting those picture’s resolution.

7.Proof-reading might be needed.

Response: Thank you for your reasonable suggestion. The proof-reading had already done.

Thanks for your valuable suggestion again. You have provided valuable suggestions for our follow-up research. We will continue relevant research to answer your questions. We hope the revision is acceptable for publication.

Sincerely,

[Yangbo Zhang]

[Hunan Agricultural University, Changsha, Hunan]

[13272032875]

[+86-0731-84635304]

[zhangyblucky@163.com]

Reviewer 2 Report

This study on jasmine tea influencing depressive-like behavior in rats is nortworthy. The authors actually concluded that jasmine tea could attenuate depressive-like behavior.

Author Response

Response to Reviewer Comments

Dear Reviewers:

Thank you for your letter and for the reviewers’ comments concerning our manuscript entitled “Jasmine Tea Attenuates Chronic Unpredictable Mild Stress-induced Depressive-Like Behavior in Rats Via the Gut-Brain Axis”. Those comments are all valuable and very helpful for revising and improving our paper. We have seriously taken the comments into consideration in revising our manuscript. Revised portion are marked in red in our manuscript. The main corrections in the paper and the responds to comments are as follows:

Reviewer #2: This study on jasmine tea influencing depressive-like behavior in rats is nortworthy. The authors actually concluded that jasmine tea could attenuate depressive-like behavior.

Response: Thanks for the positive comments and valuable suggestions. Jasmine tea is a unique variety of tea in China, which is usually made by jasmine and green tea dhool. Jasmine tea is widely loved because of the special fragrance. Jasmine tea is not a medicine, but just a healthy drink. The purpose of our research is to prove that jasmine tea can ameliorate depressive-like behaviors caused by external pressure in daily life. The results of this research showed that jasmine tea can alleviate the depressive-like behavior caused by CUMS.

Thanks for your valuable suggestion again. You have provided valuable suggestions for our follow-up research. We will continue relevant research to answer your questions. We hope the revision is acceptable for publication.

Sincerely,

[Yangbo Zhang]

[Hunan Agricultural University, Changsha, Hunan]

[13272032875]

[+86-0731-84635304]

[zhangyblucky@163.com]

This manuscript is a resubmission of an earlier submission. The following is a list of the peer review reports and author responses from that submission.

Round 1

Reviewer 1 Report

This study presents the effects of jasmine tea on depression-like behaviors, neurotransmitters, intestine and microbiota in rats forced with chronic unpredictable mild stress.  

Overall, data presentation, writing, references and English expression of this manuscript are all evaluated as inexperienced.

For example, the introduction contains a lot of information unrelated to the topic, and statistical representation of data is not appropriate.

In English writing, incorrect uses of capital and lower-case letters are found.

The authors have done a lot of experiments ad have a huge amount of data, so it is recommended to make significant revisions through logical and clear writing and data presentation.

Reviewer 2 Report

This is an interesting study that showed that  administration of Jasmine tea in a small doses can reduce depression levels and is connected to the induction of beneficial microbiota. The study is strong, however, there are few remarks.

Minor

1. Many of the figures are not clear even in large magnification

2. Many grammatical errors can be found, extensive English check could enhance the quality of the manuscript.

3. The names of the species should be in italics and without underscore

4. “PLS-DA demonstrated a clear separation between Con and Mod (Figure 5 A).” maybe the authors mean figure 6A?

Also, why did the authors choose PLS-DA instead of PCA?

5. “These findings suggested that the gut microbiota in rats was drastically reduced by CUMS-induced depression and those changes could be avoided by administrated jasmine tea.”. It is not clear if the authors are considering abundance or types of microbiota?

Major

1. I wonder if the author can explain the reason for specifically choosing to investigate 5-HT, BDNF, and GLP-1 in particular

2. I wonder if the authors can postulate/ infer why only small doses of tea can reduce depression but not higher doses.

3. It is not clear, how did the authors conclude that there is coevolution between microbiota and the host.

4. The limits of the study are not clear, for example. What are the risks of using a low dosage of tea to lower depression symptoms? can the extensive use of Jasmine tea lead to any form of dependency? Green tea can induce some form of dependency. 

If the rats were administrated jasmine tea for a while and the treatment was stopped, would that increase the depression levels to pretreatment? Would the mice show signs of dependency? Also, is the effect of Jasmine tea dependent on the age of the rats? 

Round 2

Reviewer 2 Report

I think most of the points have been addressed.  I have no further no further objections.